# END-TO-END BALANCING FOR CAUSAL CONTINUOUS TREATMENT-EFFECT ESTIMATION

## ABSTRACT

We study the problem of observational causal inference with continuous treatment. We focus on the challenge of estimating the causal response curve for infrequently-observed treatment values. We design a new algorithm based on the framework of entropy balancing which learns weights that directly maximize causal inference accuracy using end-to-end optimization. Our weights can be customized for different datasets and causal inference algorithms. We propose a new theory for consistency of entropy balancing for continuous treatments. Using synthetic and real-world data, we show that our proposed algorithm outperforms the entropy balancing in accuracy of treatment effect estimation.

## 1 INTRODUCTION

In many applications in business, social, and health sciences, we wish to infer the effect of a continuous treatment such as drug dosage or administration duration on a health outcome variable. Often, several confounding factors are common factors of influencing both treatment and response variable, therefore for accurate causal estimation of the treatment in view, we must appropriately account for their potential impact. Unlike binary treatments, causal inference with continuous treatments is largely understudied and far more challenging than binary treatments. (Galagate, 2016; Ai et al., 2021). This is primarily because continuous treatments induce uncountably many potential outcomes per unit, only one of which is observed for each unit and across units, a sparse coarsening of the underlying information needed to infer causal effects without uncertainty.

Propensity score weighting (Robins et al., 2000; Imai and Van Dyk, 2004), stand-alone or combined with regression-based models to achieve double robustness (Kennedy et al., 2017), has quickly become the state of the art for causal inference. If the weights, inversely proportional to the conditional distribution of the treatment given the confounders, are correctly modeled, the weighted population will appear to come from a randomized study. However, this approach faces several challenges: (1) The weights only balance the confounders in expectation, not necessarily in the given data (Zubizarreta et al., 2011). (2) The weights can be very large for some of units, leading to unstable estimation and uncertain inference. As a possible remedy, entropy balancing (Hainmueller, 2012) estimates the weights such that they balance confounders subject to a measure of dispersion on the weights to prevent extreme weights.

In this work, we note that low-entropy weights do not directly optimize the quality of subsequent weighted regression, and we introduce an alternative approach that does. We propose *End-to-End Balancing* (E2B) to improve the accuracy of the weighted regression used for causal inference. E2B uses end-to-end training to estimate the base weights in the entropy balancing framework. The E2B weights are thus customized for different datasets and causal inference algorithms that are based on weighting. Because we do not know the true treatment response function in real data, we propose a new approach to generate synthetic training datasets for end-to-end training.

To theoretically analyze end-to-end balancing, we define *Generalized Stable Weights* (GSW) for causal inference as a generalization of the stable weights proposed by Robins et al. (2000). We prove that weights learned by entropy balancing for continuous treatments, including E2B weights, are unbiased estimators of generalized stable weights. We also show that E2B weights are asymptotically consistent and efficient estimators of the population weights.

We perform three sets of experiments to demonstrate accuracy improvements by E2B. Two experiments with synthetic data, one with linear and another with non-linear response functions show that the E2B is more accurate than the baseline entropy balancing and inverse propensity score techniques. In the experiments on real-world data, we qualitatively evaluate the average treatment effect function learned by E2B. We also show that the base weights learned by E2B follow our intuition about up-weighting low frequency treatments.

## 2 PROBLEM DEFINITION AND RELATED WORK

**Problem Statement.** Suppose we have the triplet of $(\mathbf{x}, \mathrm{a}, \mathrm{y})$, where $\mathbf{x} \in \mathbb{X} \subset \mathbb{R}^r$, $\mathrm{a} \in \mathbb{A} \subset \mathbb{R}$ and $\mathrm{y} \in \mathbb{R}$ denote the confounders, treatments, and response variables, respectively, from an observational causal study. In our continuous treatment setting (Galagate, 2016, Ch. 1.2.6), we denote potential outcomes as $\mathrm{y}^{(a)}$, which means the value of $\mathrm{y}$ after intervention in the treatment $\mathrm{a}$ and setting its value to $a$. Given an i.i.d. sample of size $n$, $\{(\boldsymbol{x}_i, a_i, y_i)\}_{i=1}^n$, our objective is to eliminate the impact of the confounders and identify the average treatment effect function $\mu(a) = \mathbb{E}[\mathrm{y}^{(a)}]$, which is also called the response function. We make the two classic assumptions: (1) Strong ignorability: $\mathrm{y}^{(a)} \perp\!\!\!\perp \mathrm{a} \mid \mathbf{x}$. (i.e., no hidden confounders) and (2) Positivity: $0 < P(\mathrm{a}|\mathbf{x}) < 1$.

**General Causal Inference Literature.** The literature on causal inference is vast and we refer the reader to the books for the general inquiry (Pearl, 2009; Imbens and Rubin, 2015; Spirtes et al., 2000; Peters et al., 2017). Instead, we focus on reviewing the inference techniques for *continuous* treatments. In particular, we narrow down our focus on propensity score weighting approaches (Robins et al., 2000; Imai and Van Dyk, 2004), because they can either be used alone or combined with the regression algorithms to create double robust algorithms.

**Causal Inference via Weighting.** A popular approach for causal inference is to create a pseudo-population by weighting data points such that in the pseudo-population the confounders and treatments are independent. Thus, regular regression algorithms can estimate the causal response curve using the pseudo-population, which resembles data from randomized trials. Throughout this paper, we will denote the parameters of the pseudo-population with a tilde mark. Multiple forms of propensity scores have been proposed for continuous treatments (Hirano and Imbens, 2004; Imai and Van Dyk, 2004). The commonly-used *stablized weights* (Robins et al., 2000; Zhu et al., 2015) are defined as the ratio of marginal density over the conditional density of the treatments: $sw = {f(a)}/{f(a|\boldsymbol{x})}$.

**Problems with Propensity Scores.** Zubizarreta et al. (2011) list two challenges with the propensity scores: (1) The weights only balance the confounders in expectation, not necessarily in the given data. (2) The weights can be very large for some of the data points, leading to unstable estimations. The challenges are amplified in the continuous setting because computing the stabilized weights requires correctly choosing two models, one for the marginal and one for the conditional distributions of the treatments. Kang et al. (2007) and Smith and Todd (2005) provide multiple evidence that the propensity score methods can lead to large biases in the estimations. While Robins et al. (2007) propose techniques to fix the large weights problems in the binary treatment examples discussed by Kang et al. (2007), learning more accurate, bounded, and stable weights has been an active research area. Further techniques have proposed techniques to learn more robust propensity scores for binary treatments (Li et al., 2018; Zhao, 2019) too, however, the case of continuous treatments have received considerably less attention.

**Entropy Balancing.** To address the problem of extreme weights, *Entropy Balancing (EB)* (Hainmueller, 2012) estimates weights such that they balance the confounders subject to a measure of dispersion on the weights to prevent extremely large weights. Other loss functions using different dispersion metrics have been proposed for balancing (Zubizarreta, 2015; Chan et al., 2016). Zhao and Percival (2016) show that the entropy balancing is double robust. Entropy balancing has been extended to the continuous treatment setting (Fong et al., 2018; Vegetabile et al., 2021), where the balancing condition ensures that the weighted correlation between the confounders and the treatment is zero. Ai et al. (2021) propose a method for estimating the counterfactual distribution in the continuous treatment setting.

## 3 METHODOLOGY

To describe our end-to-end balancing algorithm, we first need to describe entropy balancing for continuous treatments with base weights.

### 3.1 ENTROPY BALANCING FOR CONTINUOUS TREATMENTS

**Causal Inference via Entropy Balancing.** Entropy balancing creates a pseudo-population using instance weights $w_i, i = 1, \ldots, n$, in which the treatment a and the confounders $\mathbf{x}$ are independent from each other. The independence is enforced by first selecting a set of functions on the confounders $\phi_k(\cdot) : \mathbb{X} \mapsto \mathbb{R}$, for $k = 1, \ldots, K$, that are dense and complete in $L^2$ space. Given the $\phi$ functions, we approximate the independence relationship by $\widehat{\mathbb{E}}_n[a\phi_k(\mathbf{x})] = 0$, for $k = 1, \ldots, K$, where the empirical expectation $\widehat{\mathbb{E}}_n$ is performed on the pseudo population. Hereafter, we will denote the mapped data points as $\boldsymbol{\phi}(\boldsymbol{x}_i) = [\phi_1(\boldsymbol{x}_i), \ldots, \phi_K(\boldsymbol{x}_i)]$. The $\phi_k(\cdot)$ functions can be chosen based on prior knowledge or defined by the penultimate layer of a neural network that predicts $(a, y)$ from $\mathbf{x}$. Our contributions in this paper are orthogonal to the choice of the $\phi_k(\cdot)$ functions and can benefit from ideas on learning these functions (Zeng et al., 2020). The data-driven choice of the number of bases $K$ is beyond the scope of current paper and left to future work.

**Balancing Constraint for the Continuous Treatments.** Following (Fong et al., 2018; Vegetabile et al., 2021), in the case of continuous treatments, we first de-mean the confounders $\boldsymbol{\phi}(\boldsymbol{x}_i)$ and treatments a such that without loss of generality they are taken to have mean zero. The balancing objective is to learn a set of weights $w_i, i = 1, \ldots, n$ that satisfy $\sum_{i=1}^n w_i \boldsymbol{\phi}(\boldsymbol{x}_i) = \mathbf{0}$, $\sum_{i=1}^n w_i a_i = 0$, and $\sum_{i=1}^n w_i a_i \boldsymbol{\phi}(\boldsymbol{x}_i) = \mathbf{0}$. We can write these three constraints in a compact form by defining a $(2K + 1)$–dimensional vector $\boldsymbol{g}_i = [\boldsymbol{\phi}(\boldsymbol{x}_i), a_i, a_i \boldsymbol{\phi}(\boldsymbol{x}_i)]$. The constraints become $\sum_{i=1}^n w_i \boldsymbol{g}_i = \mathbf{0}$. We stack the $\boldsymbol{g}$ vectors in a $(2K + 1) \times n$ dimensional matrix $\boldsymbol{G}$ for compact notation. In this work, without loss of generality, we will present our idea with the first order balancing, without higher order moments (Galagate, 2016; Wong and Chan, 2018; Hazlett, 2020).

**Primal and Dual EB.** A variety of dispersion metrics have been proposed as objective function for minimization such as entropy or variance of the weights (Wang and Zubizarreta, 2020). Hainmueller (2012) originally proposed minimizing the KL-divergence between the weights and a set of base weights $q_i, i = 1, \ldots, n$. Details on choice of base weights is discussed below, however, we note that $q_i = \text{const.}$ leads to minimization of the entropy of weights. Using this dispersion function and the balancing constraints, entropy balancing optimization is as follows:

$$\widehat{\boldsymbol{w}} = \operatorname*{argmin}_{\boldsymbol{w}} \ \sum_{i=1}^n w_i \log \left( \frac{w_i}{q_i} \right), \tag{1}$$

$$\text{s.t.} \qquad \text{(i) } \boldsymbol{G}\boldsymbol{w} = \mathbf{0}, \qquad \text{(ii) } \mathbf{1}^\top \boldsymbol{w} = 1, \qquad \text{(iii) } w_i \geq 0 \text{ for } i = 1, \ldots, n.$$

The above optimization problem can be solved efficiently using its Lagrangian dual:

$$\widehat{\boldsymbol{\lambda}} = \operatorname*{argmin}_{\boldsymbol{\lambda}} \ \log \left( \mathbf{1}^\top \exp \left( -\boldsymbol{\lambda}^\top \boldsymbol{G} + \boldsymbol{\ell} \right) \right), \tag{2}$$

where $\ell_i = \log q_i$ are the *log-base-weights*. Given the solution $\widehat{\boldsymbol{\lambda}}$, the balancing weights can be computed as $\boldsymbol{w} = \text{softmax}\left( -\widehat{\boldsymbol{\lambda}}^\top \boldsymbol{G} + \boldsymbol{\ell} \right)$. The softmax function is defined as $\text{softmax}(\boldsymbol{v}) = \exp \boldsymbol{v} / \left( \mathbf{1}^\top \exp \boldsymbol{v} \right)$ for any vector $\boldsymbol{v}$. The log base weight is a degree of freedom that we have in the Eq. (2) to improve the quality of causal estimation. We select the mapping dimension $K$ such that problem (2) is well-conditioned and leave the analysis of the high dimensional setting $K \approx n$ to future work. We can also add an $L_1$ penalty term to the dual objective in Eq. (2), which corresponds to approximate balancing (Wang and Zubizarreta, 2020).

In the next section we propose to parameterize the log-base-weights and learn them. Our analysis in Section 4 shows that with any arbitrary base weights, causal estimation using the weights learned in Eq. (2) will be consistent.

---

**Algorithm 1** Stochastic Training of $\ell_{\boldsymbol{\theta}}$ for End-to-End Balancing

---

**Require:** Data tuples $(\boldsymbol{x}_i, a_i, y_i)$ for $i = 1, \ldots, n$ with an unknown response function $\mu(a)$.
**Require:** Representation functions $\boldsymbol{\phi}(\cdot)$ and $\boldsymbol{\psi}(\cdot)$, split size $n_1 < n$ and batch size $B$.
 1: Generate a random set of indexes $I, |I| = n_1$ and its complement $I^c$ and split the data to $S$ and $S^c$ using them.
 2: Estimate the distribution of noise in y given $(\mathrm{a}, \mathbf{x})$ as $\widehat{F}_{\varepsilon}$.
 3: Compute $\boldsymbol{G}$ by stacking $\boldsymbol{g}_i = [\boldsymbol{\phi}(\boldsymbol{x}_i), a_i, a_i \boldsymbol{\phi}(\boldsymbol{x}_i)]$, for $i = 1, \ldots, n$.
 4: **for** Number of Iterations **do**
 5:     Generate $B$ datasets $\{(\boldsymbol{x}_i, a_i, \overline{y}_{i,b})\}_{i=1}^{n}$ for $b = 1, \ldots, B$ using $\varepsilon \sim \widehat{F}_{\varepsilon}$, and randomly selected $\overline{\mu}(a)_b$ response functions.
 6:     $\ell_i \leftarrow \ell_{\boldsymbol{\theta}}(\boldsymbol{\psi}(a_i, \boldsymbol{x}_i))$.
 7:     $\widehat{\boldsymbol{\lambda}} \leftarrow \operatorname{argmin}_{\boldsymbol{\lambda}} \left\{ \log\left(\mathbf{1}^{\top} \exp\left(-\boldsymbol{\lambda}^{\top} \boldsymbol{G} + \boldsymbol{\ell}\right)\right) \right\}$ using only $S$ data.
 8:     $\boldsymbol{w} \leftarrow \operatorname{softmax}\left(-\widehat{\boldsymbol{\lambda}}^{\top} \boldsymbol{G} + \boldsymbol{\ell}\right)$ using only $S^c$ data.
 9:     $\widehat{\mu}(a)_b \leftarrow$ weighting-based causal estimates using $(a_i, \overline{y}_{i,b}, w_i)$ in $S^c$ for $b = 1, \ldots, B$.
10:     Take a step in $\boldsymbol{\theta}$ to minimize $\frac{1}{B} \sum_{b=1}^{B} \left(\widehat{\mu}(a)_b - \overline{\mu}(a)_b\right)^2$.
11: **end for**
12: **return** The $\ell_{\widehat{\boldsymbol{\theta}}}$ function.

---

## 3.2 Learning the Base Weights

Hainmueller (2012) suggests two approaches for choosing base weights: (1) weights obtained from a conventional propensity score model and (2), in the context of survey design, using knowledge about the sampling design. We argue that a data-driven approach that learns customized base weights for each pair of dataset and weighted causal regression algorithm can further improve performance.

To address this problem, we define the log-base-weights $\ell$ as a parametric function (e.g., a neural network) of the treatment variable; i.e. $\ell_{\boldsymbol{\theta}}(\cdot)$. We learn the base weights with the goal of improving the accuracy of the subsequent weighted regression. This is challenging because simply optimizing the weighted regression loss (e.g., weighted MSE) leads to degenerate results. That is, learning $\ell$ to minimize the regression loss will lead to exclusion of the difficult-to-predict data points from the regression, which is undesirable. Thus, we need to find another loss function to optimize, ideally a loss function that directly minimizes the error in estimation of the response function $\mu(a)$.

Our idea for learning the parameters of the base weights is to generate multiple pseudo-responses $\overline{y}$ with randomly generated response functions $\overline{\mu}(a)$. Now that we know the true response function $\overline{\mu}(a)$ in the randomly generated data, we can perform causal inference and obtain the estimation of the known response curve $\widehat{\mu}(a)$ using our weights. Algorithm 1 outlines our stochastic training of the base weight function. First, in Step 2, we estimate the distribution of noise using the residuals of regressing y over $(\mathrm{a}, \mathbf{x})$, capturing the possible heteroskedasticity in the noise. Then, in each iteration, we draw a batch of possible datasets. To generate each dataset, we randomly choose a response function $\overline{\mu}(a)$ and use it to generate the entire dataset (see Section 5.1 for examples of random functions). For the entire batch, we use $\ell_{\boldsymbol{\theta}}$ to learn the log-base-weights, and subsequently learn the weights in lines 7–8. In line 9 we use a weighted regression algorithm to find our estimation $\widehat{\mu}(a)$ of the randomly-generated $\overline{\mu}(a)$. Our loss function is the mean squared error between the latter quantities. While we call our algorithm *End-to-End Balancing* (E2B) because of our end-to-end optimization.

**Sample Splitting.** The E2B procedure involves estimation of two sets of parameters $\boldsymbol{\theta}$ in the $\ell_{\boldsymbol{\theta}}$ and $\boldsymbol{\lambda}$ for entropy balancing. The joint estimation of $\boldsymbol{\theta}, \boldsymbol{\lambda}$ on a single sample will result in bias (Chernozhukov et al., 2018). Thus, we split the sample to two mutually exclusive parts and perform the optimizations on separate partitions of data.

**Choice of Random Response Functions.** Ideally, we should rely on domain experts for choosing the random set of response functions $\overline{\mu}(a)$ that includes the true response function. Alternatively, we can choose broad function classes such as random piecewise smooth functions or polynomial

functions with random coefficients. We can also use generative adversarial networks to generate data that is more similar to our sample (Athey et al., 2021).

**Features Fed to $\ell_{\boldsymbol{\theta}}$.** We can feed the raw values of the treatments and any handcrafted features, denoted by $\boldsymbol{\psi}(a_i, \boldsymbol{x}_i)$. We empirically find that $\boldsymbol{\psi}(a_i, \boldsymbol{x}_i) = (\log p(a_i), \log p(a_i|\boldsymbol{x}_i))$ makes training the $\ell_{\boldsymbol{\theta}}$ easier. We describe the details of our neural network model for $\ell_{\boldsymbol{\theta}}$ and our techniques for training in Appendix B.

**Weighted Regression Algorithms.** To be able to differentiate the loss function with respect to $\boldsymbol{\theta}$, we need weighted regression algorithms whose estimates are differentiable with respect to the weights. In the linear average treatment effect function we choose weighted linear regression and in the non-linear setting we use the weighted polynomial regression and the local kernel regression, as used by Flores et al. (2012).

**Double Robustness.** Zhao and Percival (2016) show that in the binary case, the entropy balancing is double robust. We do not attempt to show double robustness for E2B because we see E2B as a meta algorithm that learns customized weights for each dataset and algorithm. We can either (1) plug-in the E2B weights in the double robust algorithm and expect improved accuracy, or (2) learn weights that directly minimize the error of double robust algorithms such as (Kennedy et al., 2017).

## 4 ANALYSIS

We prove that for any arbitrary choice of the log-base-weight function $\ell_{\boldsymbol{\theta}}$, our approach consistently estimates causal effects. Before proving the consistency results, we characterize the quantity that our solution converges to. All long proofs are relegated to Appendix A.

**Definition 1.** *Generalized Stable Weights. Suppose $f(a, \boldsymbol{x})$ denote the joint probability density function of treatments and confounders in a population. Suppose $\widetilde{f}(a)$ and $\widetilde{f}(\boldsymbol{x})$ denote two arbitrary density functions, possibly different with the marginal density functions in our population, that satisfy $\mathbb{E}_{\mathbf{x}\sim\widetilde{f}(\boldsymbol{x})}[\mathbf{x}] = \mathbf{0}$ and $\mathbb{E}_{\mathbf{a}\sim\widetilde{f}(a)}[\mathbf{a}] = 0$. We define the Generalized Stable Weights as follows*

$$w_{GSW}(a, \boldsymbol{x}) = \frac{\widetilde{f}(a)\widetilde{f}(\boldsymbol{x})}{f(a, \boldsymbol{x})}. \tag{3}$$

**Remark.** Our definition generalizes the stabilized weights defined by Robins et al. (2000), where $\widetilde{f}(a)$ and $\widetilde{f}(\boldsymbol{x})$ match the marginal probability density functions in the original population.

**Proposition 1.** *The generalized stable weights $w_{GSW}$ satisfy $\mathbb{E}\left[w_{GSW}\mathbf{a}\mathbf{x}\right] = \mathbf{0}$.*

*Proof.*

$$\mathbb{E}\left[\mathbf{w}_{GSW}\mathbf{a}\mathbf{x}\right] = \mathbb{E}\left[\frac{\widetilde{f}(\mathbf{a})\widetilde{f}(\mathbf{x})}{f(\mathbf{a}, \mathbf{x})}\mathbf{a}\mathbf{x}\right] = \int\int\frac{\widetilde{f}(a)\widetilde{f}(\boldsymbol{x})}{f(a, \boldsymbol{x})}a\boldsymbol{x}\,\mathrm{d}F_{\mathbf{a},\mathbf{x}}(a, \boldsymbol{x}) = \int a\widetilde{f}(a)\mathrm{d}a\int\boldsymbol{x}\widetilde{f}(\boldsymbol{x})\mathrm{d}\boldsymbol{x} = \mathbf{0},$$

where the last equation is because of zero mean assumption for the $\widetilde{f}(a)$ and $\widetilde{f}(\boldsymbol{x})$ distributions. $\square$

Now, we can show that with an appropriate choice of the $\phi$ functions, the solution of Eq. (2) approximates the generalized stable weights. Consider the population version of Eq. (2):

$$\boldsymbol{\lambda}^{\star} = \underset{\boldsymbol{\lambda}}{\arg\min}\log\left(\mathbb{E}\left[\exp(\mathbf{g}^{\top}\boldsymbol{\lambda} + \ell)\right]\right). \tag{4}$$

The weights corresponding to $\boldsymbol{\lambda}^{\star}$ can be calculated as $w^{\star} = C\exp(\mathbf{g}^{\top}\boldsymbol{\lambda}^{\star} + \ell)$, where $C = \left(\int\exp(\mathbf{g}^{\top}\boldsymbol{\lambda}^{\star} + \ell)\mathrm{d}F(a, \boldsymbol{x})\right)^{-1}$ is the normalization constant.

**Assumptions.**

1. $f(a, \boldsymbol{x}) \geq c > 0$ for all $(a, \boldsymbol{x}) \in \mathbb{A} \times \mathbb{X}$ pairs, where $c$ is a constant.

2. Suppose the basis functions are dense and rich enough such for some small values of $\delta_{\boldsymbol{\phi}_K}$ that they satisfy:
$$\mathbb{E}[\mathrm{a}\boldsymbol{\phi}(\mathbf{x})] = \mathbf{0} \quad \text{only if} \quad \sup_{a,\boldsymbol{x}} |f(a, \boldsymbol{x}) - f(a)f(\boldsymbol{x})| = \delta_{\boldsymbol{\phi}_K}.$$

3. Suppose the population problem in Eq. (4) has a unique solution $\boldsymbol{\lambda}^\star$ and the corresponding weights are denoted by $w^\star$.

The following theorem shows that the solution to Eq. (4) converges to $w_{GSW}$:

**Theorem 1.** *Given the assumptions, the solution to the population problem satisfies:*
$$\sup_{a,\boldsymbol{x}} |w^\star(a, \boldsymbol{x}) - w_{GSW}(a, \boldsymbol{x})| \leq \delta_{\boldsymbol{\phi}_K}/c. \tag{5}$$

If we select the function set $\boldsymbol{\phi}_K$ such that $\delta_{\boldsymbol{\phi}_K} = o(1)$, the theorem shows that $w^\star(a, \boldsymbol{x})$ is an unbiased estimator of $w_{GSW}(a, \boldsymbol{x})$. Notice that Assumption 1 is only slightly stronger than the common positivity assumption. Assumption 2 requires us to select the mapping functions such that zero the correlation between the mapped confounders and the treatment implies their independence. We provide the proof in Appendix A.1.

Note that the quality of the $\psi$ features and neural network training does not affect the unbiasedness of the E2B because of the balancing constraint is still satisfied. The flexibility in choice of $\widetilde{f}$ distributions in the definition of $w_{GSW}$ is due to the fact that we require only first order balancing. If we enforce higher order balancing constraints in the form of $\mathbb{E}[w^\star\phi_1(\mathrm{a})\phi_2(\mathbf{x})] = \mathbb{E}[\phi_1(\mathrm{a})] \cdot \mathbb{E}[\phi_2(\mathbf{x})]$ for any suitable functions $\phi_1$ and $\phi_2$, Theorem 1 in (Ai et al., 2021) shows that $w^\star = {}^{f(a)}/_{f(a|\boldsymbol{x})}$. The more flexible form of weights in Eq. (3) allows us to pick the marginals $\widetilde{f}(a)$ and $\widetilde{f}(\boldsymbol{x})$ with more freedom. In this work, we have chosen a data-driven way to learn them.

Finally, the following theorem establishes the asymptotic consistency and normality result for each individual weight estimated by E2B, under the common regularity conditions for problem (2).

**Theorem 2.** *Suppose $\Lambda \subset \mathbb{R}^{2K+1}$ is an open subset of Euclidean space and the solution $\widehat{\boldsymbol{\lambda}}_n \in \Lambda$ to Eq. (2) is within the subset. The weights estimated by Eq. (2) are asymptotically normal for $i = 1, \ldots, n$:*
$$\sqrt{n}\left(\widehat{w}_n(a_i, \boldsymbol{x}_i) - w^\star(a_i, \boldsymbol{x}_i)\right) \xrightarrow{d} \mathcal{N}(0, \sigma^2(a_i, \boldsymbol{x}_i)). \tag{6}$$

*We provide the population form of $\sigma^2(a_i, \boldsymbol{x}_i)$ and an unbiased sample estimate for it in Appendix A.2.*

## 5 EXPERIMENTS

We use two synthetic and one real-world datasets to show that E2B outperforms the baselines. In the synthetic datasets, we have access to the true treatment effects; thus we measure accuracy of the algorithms in recovering the treatment effects. In the real-world data, we qualitatively evaluate the estimated causal treatment effect curve and inspect the learned log-base-weight function.

**Baselines.** A key baseline in our study is the Inverse Propensity score Weighting (IPW) with Stable Weights (Robins et al., 2000) as the most commonly used technique. To avoid extreme weights and prevent instability, we trim (Winsorize) the weights by $[5, 95]$ percentiles (Cole and Hernán, 2008; Chernozhukov et al., 2018). However, the main baseline in our experiments is Entropy Balancing (Vegetabile et al., 2021), which is equal to E2B with $\ell_{\boldsymbol{\theta}} = \text{const}$, corresponding to the constant base weights. EB allows us to do an ablation study and see the exact amount of improvement by learning a customized $\ell_{\boldsymbol{\theta}}$ function. We also include EB with the stabilized weights (SW) as base weights ($\ell_{\boldsymbol{\theta}} = \log \widehat{p}(a) - \log \widehat{p}(a|\boldsymbol{x})$). Finally, we also include the permutation weighting algorithm (Arbour et al., 2021) that proposes to compute the weights using permutation of the treatments and a classifier that predict the probability of being permuted. We provide further details on this algorithm in Appendix B.4.

Table 1: Average RMSE for estimation of the response functions. The results are in the form of "mean (std. err.)" from 100 runs.

| Algorithm | Linear | Non-linear |
|---|---|---|
| Inverse Propensity Weighting (SW) | 2.057 (0.437) | 0.530 (0.025) |
| Permutation Weighting | 1.1543 (6.580) | 0.525 (0.250) |
| Entropy Balancing (Const.) | 0.880 (0.072) | 0.335 (0.022) |
| Entropy Balancing (SW) | 0.652 (0.059) | 0.403 (0.025) |
| End-to-End Balancing | **0.383** (**0.035**) | **0.276** (**0.014**) |

**Training Details.** We provide the details of the neural networks used for the $\ell_{\boldsymbol{\theta}}$ and propensity score estimation for IPW in Appendix B. All neural networks are trained using Adam (Kingma and Ba, 2014) with early stopping based on validation error. The learning rate and architectural parameters of the neural networks are tuned via hyperparameter search on the validation data.

## 5.1 SYNTHETIC DATA EXPERIMENTS

**Linear.** We use the following steps to generate 100 datasets, each with 1000 data points.

1. Generate confounders $\mathbf{x} \in \mathbb{R}^5$, $\mathbf{x} \sim \mathcal{N}(\mathbf{0}, \boldsymbol{\Sigma})$, where $\boldsymbol{\Sigma}$ is a tridiagonal covariance matrix with diagonal and off-diagonal elements equal to $1.0$ and $0.2$, respectively.

2. $a \sim \mathcal{N}(\mu_a, 0.3^2)$, where $\mu_a = \sin(\boldsymbol{\beta}_{xa}^\top \mathbf{x})$ and $\beta_{xa,k} \sim \text{Unif}(-1, 1)$ for $k = 1, \ldots, 5$.

3. $y \sim \mathcal{N}(\mu_y, 0.5^2)$, where $\mu_y = \boldsymbol{\beta}_{xy}^\top \mathbf{x} + \beta_{ay} a$, where $\beta_{ax}, \beta_{xy,k} \sim \mathcal{N}(0, 1)$ for $k = 1, \ldots, 5$.

We use weighted least squares as the regression algorithm and report the average $|\widehat{\beta_{ay}} - \beta_{ay}|$ over all 100 datasets.

**Nonlinear** We first generate confounders $\mathbf{x}$ and treatments $a$ similar to steps 1 and 2 of the linear case. Then, we generate the response variable according to $y \sim \mathcal{N}(\mu_y, 0.5^2)$, where $\mu_y = \boldsymbol{\beta}_{xy}^\top \mathbf{x} + h_{\boldsymbol{\gamma}_{ay}}(a)$, where $\beta_{xy,k} \sim \mathcal{N}(0, 1)$ for $k = 1, \ldots, 5$. The Hermit polynomials are defined as $h_{\boldsymbol{\gamma}}(z) = \gamma_0 + \gamma_1 z + \gamma_2(z^2 - 1) + \gamma_3(x^3 - 3x)$. Similar to the linear case, we generate 100 samples of size 1000. We use the weighted polynomial regression as the regression algorithm to estimate $\widehat{\boldsymbol{\gamma}}$ and report the average RMSE between true $\boldsymbol{\gamma}$ and $\widehat{\boldsymbol{\gamma}}$. We report the mean and standard error of errors on 100 datasets in Table 1.

As seen in Table 1, in both linear and non-linear datasets, the E2B is significantly more accurate in uncovering the true treatment response functions. Both constant and IPW base weights perform worse than the base-weights learned by end-to-end balancing. As Robins et al. (2007) caution, synthetic data evaluation might exacerbate the extreme weights issues because unlike real data, usually no manual inspection of weights are done.

To gain more insights, in Figure 1, we plot the log-base weight function that we learn as a function of $\log(\widehat{p}(a))$ and $\log(\widehat{p}(a|\mathbf{x}))$. We align all curves at their starting point and plot the median of 100 runs. Both figures, show more variations in the $\log(\widehat{p}(a))$–axis, rather than the $\log(\widehat{p}(a|\mathbf{x}))$–axis. Not that, especially in the linear case, the smaller conditional probability leads to larger base-weights, inline with the IPW base-weights. Finally, the complexity of the plots emphasizes the need for end-to-end methods for learning weights.

## 5.2 REAL DATA EXPERIMENTS

We study the impact of $PM_{2.5}$ particle level on the cardiovascular mortality rate (CMR) in 2132 counties in the US using the data provided by the National Studies on Air Pollution and Health (Rappold, 2020). The data is publicly available under U.S. Public Domain license. The $PM_{2.5}$ particle level and the mortality rate are measured by $\mu g/m^3$ and the number of annual deaths due to cardiovascular conditions per 100,000 people, respectively. We use only the data for 2010 to simplify the experiment setup; thus we measure the same year impact of $PM_{2.5}$ particle level. Other than

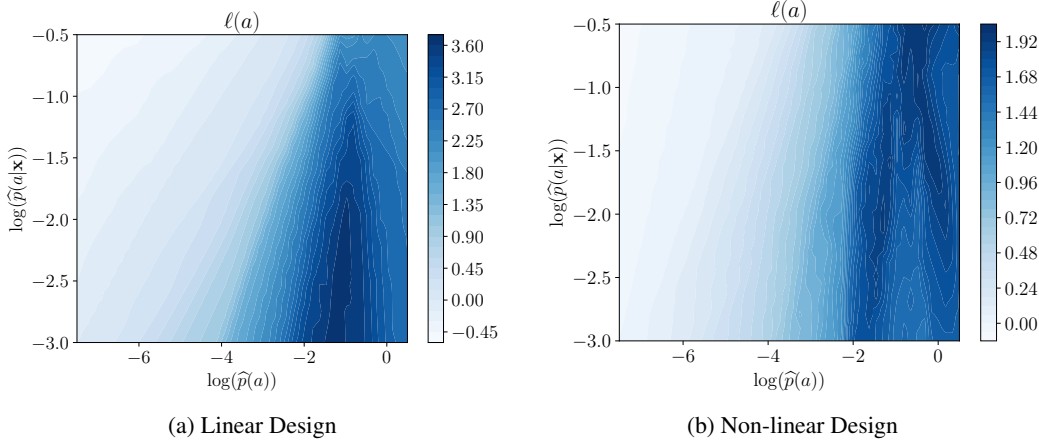

(a) Linear Design

(b) Non-linear Design

Figure 1: The estimated log-base-weight function $\ell_{\boldsymbol{\theta}}$ as a function of logarithms of the empirical density of the treatment $\log(\widehat{p}(a))$ and conditional distribution $\log(\widehat{p}(a|\mathbf{x}))$. We perform the experiment 100 times and report the median and the inter-quantile range. We align all curves by normalizing their value at the beginning to zero.

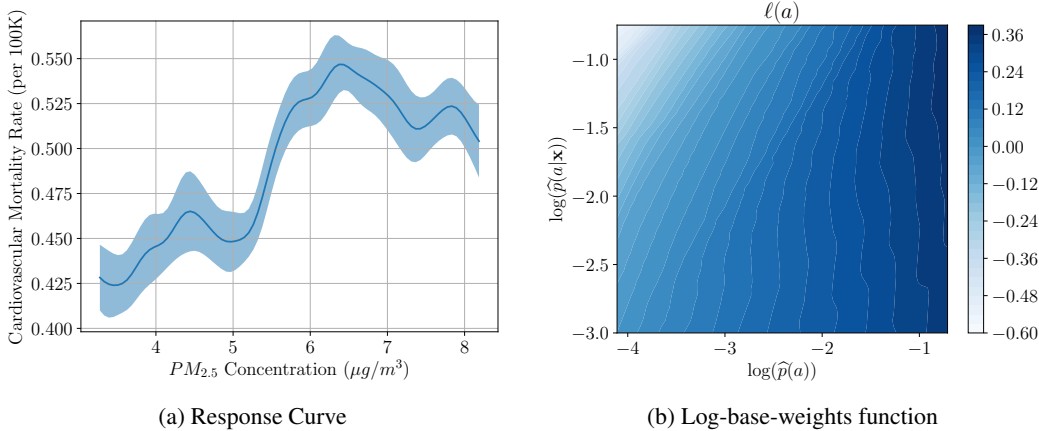

(a) Response Curve

(b) Log-base-weights function

Figure 2: (a) The average treatment effect curve for measuring the impact of $PM_{2.5}$ concentration on the cardiovascular mortality rate. We perform the experiment 100 times and report the mean and $\pm$std range. (b) The estimated log-base-weight function $\ell_{\boldsymbol{\theta}}$ as a function of logarithm of the empirical density of the treatment $\log(\widehat{p}(a))$.

the treatment and response variables, the data includes 10 variables such as poverty rate, population, and household income, which we use as confounders. We provide the descriptive statistics and the histograms for the treatment and effect in Appendix C.

To train E2B, we create the random dataset (Line 5 in Algorithm 1) using Hermite polynomials of max degree 3, $\mu_y = \left| h_{\boldsymbol{\gamma}_{xy}}(\boldsymbol{\beta}_{xy}^{\top}\mathbf{x}/\|\boldsymbol{\beta}_{xy}^{\top}\mathbf{x}\|_2) + h_{\boldsymbol{\gamma}_{ay}}(a) \right|$. We use absolute value to capture the positivity of our response variable. The data also shows heteroskedasticity; we model the noise as a zero mean Gaussian variable with variance $\sigma^2(\widehat{y}) = 6.00\widehat{y}$. For regression, we use the non-parametric local kernel regression algorithm. We measure the uncertainty in the curves using the deep ensembles technique (Lakshminarayanan et al., 2017) with 100 random ensembles.

Figure 2a shows the average treatment effect curve for the impact of $PM_{2.5}$ on CMR. We show the one standard deviation interval using the shaded areas. Starting around $PM_{2.5} = 5.3\mu g/m^3$ the curve increases with a steep slope; confirming the previous studies that increased $PM_{2.5}$ levels increase the probability of cardiovascular mortality. Our results are generally aligned with the results reported in (Wu et al., 2020). We can see that after $PM_{2.5} = 6.4\mu g/m^3$ the curve plateaus and

mortality rate stays at elevated levels. Looking at the histogram of the treatments in Figure 3a in the appendix, we observed that most counties have $PM_{2.5}$ between 6 and 8. This might justify the fluctuations that we see in this interval and may allude about potential unmeasured confounders.

Figure 2b shows the log-base-weight function that we learn in this data. Similar to the synthetic experiments, we show the median of 100 runs. While the plot shows smaller variations, it is generally inline with the observations we had in the synthetic data.

## 6 DISCUSSION

Causal inference is a well-studied problem; its main goal is to remove biases due to confounding by balancing the population to look similar to randomized controlled trials. Removing the impact of confounders can play a critical role in reducing and possibly eliminating bias in our decision making leading to potentially positive societal impacts. Our results rely on two classical assumptions: (1) unconfoundedness and (2) positivity. While these assumptions are sometimes reasonable in practice, their violations might lead to biased causal inferences. For example, the positivity assumption might be violated if we do not collect any data for a sub-population. Overall, the debiasing property of causal inference should not relieve us from rigorous data collection and analysis setup. In our experiments, we have been careful to quantify uncertainty in our causal estimation and be wary of over-confidence in our results. We performed our experiments on a CPU machine with 16 cores from a cloud provider that uses hydroelectric power.

## 7 CONCLUSION

We observed that in the entropy balancing framework, the base weights provide an extra degree of freedom to optimize the accuracy of causal inference. We propose end-to-end balancing (E2B) as a technique to learn the base weight such that they directly improve the accuracy of causal inference using end-to-end optimization. In our theoretical analysis we find the quantity that E2B weights are approximating and discuss E2B's statistical consistency. Using synthetic and real-world data, we show that our proposed algorithm outperforms the entropy balancing in terms of causal inference accuracy.

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
