# OpenReview forum: "End-to-End Balancing for Causal Continuous Treatment-Effect Estimation"
_ICLR.cc/2022/Conference — ICLR 2022 Submitted_

### Official Review · Reviewer_UC4j · 2021-10-29

**Correctness:** 4
**Technical Novelty And Significance:** 4
**Empirical Novelty And Significance:** 4
**Recommendation:** 8
**Confidence:** 4

**Main Review:**

This is an excellent paper that identifies a specific problem - weighting for estimating a continuous treatment effect in observational causal settings - and builds on existing work to propose a thorough solution. The authors identify that propensity weighting can be unstable in specific instances (it only balances in expectation, and can lead to unstable weights), and that entropy balancing weights do not directly optimize for treatment effect estimation. Addressing those problems, they suggest End-to-End Balancing (E2B), which does optimize for the subsequent regression step and which provides asymptotically consistent estimators of population weights.

Strong points: The paper is technically correct and thorough. The experimental setup is reasonable, including both simulation studies and real data experiments. When evaluating competitors, the authors trim the inverse probability weights to give competing methods the best possible performance in simulations, also comparing to standard entropy balancing estimates. The ideas are novel and overall well-presented.

Weak points: There are very few weak points with this paper. My only suggestion would be to increase discussions of propensity weights and entropy balancing for context; many readers will be familiar with one more than the other, since this paper lies at the intersection of causal inference coming from more economics/mathematical and computer science backgrounds. (This is a great thing!)

**Summary Of The Paper:**

This paper introduces an estimation method, End to End Balancing (E2B), to improve propensity weighted regression estimates for continuous treatments in observational causal settings. Propensity scores are a popular and successful tool to help us analyze observational studies, but suffer from two major problems: they only balance confounders in expectation, and they can be large and unstable. Entropy balancing has been used before to estimate weights that balance confounders by creating a pseudo-population in which treatment and confounders are independent. This method, however, does not optimize for the regression we perform afterwards, and hence misses an optimization opportunity. The authors close this gap by leveraging the entropy balancing framework to find base weights that do optimize for the treatment effect estimation regression in the continuous treatment setting. The paper includes proof of asymptotic consistency and experimental validations.

**Summary Of The Review:**

I think this is a very strong paper and recommend it for acceptance. Methods to learn continuous treatment effects will be of broad interest across causal inference researchers and practitioners, and the authors back up a clever idea with asymptotic theory and empirical validations. I enjoyed reading it and I think many others would too!

---

### Official Review · Reviewer_gT3S · 2021-10-31

**Correctness:** 3
**Technical Novelty And Significance:** 4
**Empirical Novelty And Significance:** 1
**Recommendation:** 6
**Confidence:** 4

**Main Review:**

Causal inference with continuous treatments is a hard problem and I believe the authors propose a nice method for improving on the commonly used method of entropy balancing. The approach is nice in that it is compatible with other advances in the general entropy balancing technique, such as better choice of basis functions.

Below, I list some questions and suggestions.

1. It would be good to state early on that you are dealing with observational data. Similarly, though they come up in the discussion, it would be good to state the associated assumptions of ignorability and positivity early on.
2. "Causal inference accuracy" is not a well defined term to the best of my knowledge. It might be good to be more explicit in stating that you are interested in accuracy of treatment effect estimation.
3. Related to the above, can you say anything about your estimand of interest? I believe this would have more to do with entropy balancing than your specific procedure, but it would still be good to mention whether you're typically interested in conditional or average treatment effects.
4. Can you clarify the comment on double robustness? Is there any notion of misspecification of the weights? Typically double robustness in causal refers to being able to misspecify an outcome model or a treatment selection model and still be consistent -- how does that play in here?
5. “We note that the choice of uniform distribution for qis, leads to minimization of the entropy of weights”. To make sure I understand, do you mean that minimizing the KL between uniformly chosen base weights and the weights yields weights whose entropy is minimized?
6. What is \ell(a) on the top axis of Figure 1 and how does it differ from the bottom axis?
7. I am not sure I understand Figure 1. What is varying across the x - axis given that it is labeled with a distribution -- is it just a? Similarly for the y axis -- what is x fixed at? What is the implication of the variation you cite? Does the variation even mean anything, if all the quantities on the axes are just estimated?
8. In step 5 of the algorithm, do you simply generate error B different times or do you also choose new response functions each time so that a new random response function is chosen for each b = 1, ..., B and also for each iteration?
9. I want to make sure I understand the response functions. Say you choose the set of polynomials of max degree 3. Then when you choose a 'random function' this just refers to the degree of the polynomial? How are the coefficients chosen?
10. I'm confused why / what it means that your theory is independent of the choice of the pseudo-response generating functions, which seem to be a critical aspect of your set-up. The theory seems to suggest that I could choose e.g. constant functions and be alright. Is there any gain in efficiency if I properly specify the outcome model?
11. What does it mean to look at the median/IQR in the real world example; that is, what are you averaging over? Different NN initializations? Are you averaging over the same thing (and then averaging over different datasets) in the synthetic experiments?
12. What response functions do you use for the synthetic experiments? It would be nice to see the effect of some sort of misspecification.

**Summary Of The Paper:**

The authors propose a way to intelligently choose the base weights in an entropy balancing framework for estimating treatment effects from observational data. Typically, the weights are chosen arbitrarily (e.g. to be uniform) or based off of domain expertise, which is not always available. The weights are chosen by positing forms for the response function, generating pseudo-responses, and then finding weights that maximize treatment effect estimation accuracy. In this sense, the procedure is end-to-end because the weighting is done specifically with the subsequent estimation in mind. The authors provide theoretical results showing that the weights are asymptotically normal and that they converge to the population minimizers of the entropy balancing optimization problem. Experimental results on synthetic datasets show that the proposed method is superior to entropy balancing with naive weights with respect to treatment effect estimation accuracy.

**Summary Of The Review:**

I think this is an interesting, novel idea with potential for empirical benefits. However, I am wondering about the choice of response functions, both empirically (how does misspecification affect the procedure?) and theoretically (why is the theory independent of this aspect of the procedure?). I am intrigued by the idea but will adjust my score based off of the author responses.

---

### Official Review · Reviewer_E7bS · 2021-11-03

**Correctness:** 4
**Technical Novelty And Significance:** 3
**Empirical Novelty And Significance:** 3
**Recommendation:** 8
**Confidence:** 3

**Main Review:**


Thanks for the authors for an interesting work showing that one can improve causal inference by randomly generated outcomes. This is a simple and effective way to improve balancing of weights. The contributions in algorithms, analysis, and empirical evaluation seem balanced (no pun intended.) Yet I still see some possible improvements.

You didn’t explain psi before introducing algorithm and casually explained in “Features Fed to ell“. Can I just think of it as any proper mapping used before applying \ell? One can think of \ell_\theta embedding \psi (e.g., representation learning and output layer… something you have described for phi in paragraph “Causal Inference via Entropy Balancing” in Sec 3.1) Should \psi be explicit as requirement or can it be learned as well?

Is w_GSW irrelevant to the random sampling of response function? Meaning that, the analysis is rather about Section 3.1 (existing methods) but not about 3.2 (new algorithmic contribution)?

“We empirically find that (ai, xi) = (log p(ai), log p(ai|xi)) makes training the \ell easier.” It would be helpful for readers to explain it one or two sentence, any intuition behind it?

Can you check whether the learned base weight is somewhere in-between Uniform and SW? Would you be able to add an experiment EB with base weight as the convex combination of SW and Uniform (alpha SW+ (1-alpha) Unif) for some 0<alpha<1 so that we can better make sense of the learned base weight is better than base weights based on some, simple convex combination of SW and Uniform. Further, what would be the alpha that results in the weights closest to the learned weights? (for both linear and non-linear setting)?

I would like to see the result with or without sample splitting.

“We can also use generative adversarial networks to generate data that is more similar to our sample.“  Did you try the idea and didn’t it work well or the idea is under development? When I first read through the paper, I thought about GAN could be a better option than random ones. Then, Voila, there is the statement.

Minor,
bold \ell for Eq (4) and a subsequent expression.



**Summary Of The Paper:**

This paper propose a method to estimate causal effect when the treatment is continuous variable. In particular, the authors focused on employing entropy balancing (EB) to find weights that are not to extreme in performing weighted regression to reduce the variance of the estimate. More specifically, the authors proposed to an end-to-end approach to learn the “base weight” to be used in EB in contrast to the original, uniform distribution using randomly samples.


**Summary Of The Review:**

Overall liked the idea but the novelty can be limited. Yet it improves the quality of estimator substantially. Hence, I recommend acceptance.

---

### Official Review · Reviewer_G9aV · 2021-11-03

**Correctness:** 3
**Technical Novelty And Significance:** 3
**Empirical Novelty And Significance:** 2
**Recommendation:** 3
**Confidence:** 4

**Main Review:**

This paper has a promising premise: the algorithm proposed is very interesting and appears to hold a lot of promise. Unfortunately, the paper lacks some connection to the wider literature and experimental evidence that prevents me from recommending acceptance at this time. In particular:

(1) At this point there is a fairly robust literature on balancing weights beyond entropy balancing. Below are a sampling of references. The amount of time specifically citing the shortcomings of entropy balancing itself appears almost a distraction–many of the papers below have empirical performance that far exceed entropy balancing and have strong theoretical properties. It would be helpful if the authors addressed this literature and include at least some of them in experimental comparisons.

Hazlett, C. (2020). Kernel Balancing: A flexible non-parametric weighting procedure for
estimating causal effects. Statistica Sinica, 30, 1155–1189.

Li, F., Morgan, K. L., & Zaslavsky, A. M. (2018). Balancing covariates via propensity score
weighting. Journal of the American Statistical Association, 113 (521), 390–400.

Tan, Z. (2020). Regularized calibrated estimation of propensity scores with model misspecification and high-dimensional data. Biometrika, 107 (1), 137–158.

Wong, R. K., & Chan, K. C. G. (2018). Kernel-based covariate functional balancing for
observational studies. Biometrika, 105 (1), 199–213.

Zhao, Q. (2019). Covariate balancing propensity score by tailored loss functions. Annals of
Statistics, in press.

Zheng, W., & van der Laan, M. J. (2011). Cross-validated targeted minimum-loss-based
estimation. Targeted learning (pp. 459–474). Springer.

Zubizarreta, J. R. (2015). Stable weights that balance covariates for estimation with incomplete outcome data. Journal of the American Statistical Association, 110 (511), 910–922.

(2) There is also a growing literature on balancing weights for continuous treatments. Again, I've listed some publications below. Arbour, et al.  also introduce the idea of targeting the stabilized weight directly (i.e., the product distribution) via density ratio estimation.

Kallus, Nathan, and Michele Santacatterina. "Kernel optimal orthogonality weighting: A balancing approach to estimating effects of continuous treatments." arXiv preprint arXiv:1910.11972 (2019).

Arbour, D., Dimmery, D., & Sondhi, A. (2021) Permutation Weighting. ICML.

Vegetabile, Brian G. et al. “Nonparametric estimation of population average dose-response curves using entropy balancing weights for continuous exposures.” Health Services and Outcomes Research Methodology 21 (2021): 69-110.

(3) There are a few other examples within the literature which combine weighting and outcome modeling more closely including TMLE, Johansson, Shalit & Sontag (2016), and Shi, Blei & Veitch (2019). In addition in "Adaptive normalization for IPW estimation", Khan and Ugander provide a connection between self-normalization and an implicit regression adjustment.

(4) It would be nice if the authors included more challenging empirical evaluations. The data generating mechanisms are very simple and are unlikely to match what practitioners are likely to see in practice. As I mentioned above, it would also be great to see evaluation against current methods. More importantly though, it would be very useful to see some ablation studies to see the effect of each constituent part of the algorithm, e.g. the pseudo outcome, weighted regression, etc. to (a) have a sense of what accounts for performance and (b) have a sense of the limitations of the proposed approach.


**Summary Of The Paper:**

The authors propose an alternative to balancing weights that creates a series of pseudo-outcomes via a procedure similar to the wild bootstrap, estimating balancing weights, and then updating the causal parameter. The authors provide some theoretical evidence and a set of synthetic experiments to assess their claims.

**Summary Of The Review:**

I think this is a very interesting idea, but unfortunately I'm not quite ready to recommend acceptance. My reluctance is largely due to two factors (1) a lack of connection to the broader literature, and (2) the need for more comprehensive experimental evidence exploring the performance of the proposed method. After reading the authors' responses, I have strengthened my score. As I detail below, I do not believe that the current paper sufficiently engages with prior art (especially in the experiments, but also in the main text), and as a result it is difficult to properly discern the relative contribution of this work.

---

### Decision · Program_Chairs · 2022-01-20

**Decision:**

Reject

**Comment:**

The paper provides a new way of weighting data to build weighted estimators of causal effects (which themselves can be used in other contexts, e.g. doubly-robust estimators). It's novel in the sense that it optimizes the choice of weighting based on information about the response function space. The approach is simple to implement, and opens up other possibilities for different classes of estimators.

I liked it. I think the paper is nearly there in terms of a well-rounded contribution. But I have to say that I did share the concern about the choice of random response functions. It's not only a matter of function space (everybody wants the most flexible one), but also of the random measure that goes in it - so the more flexible the random space, the least understood (to me at least) is the influence of the random measure. Surely that are choices of function space distributions that can do worse than uniform weights for some classes of problems? It's not that it's a implausible starting point (Bayesians do it all the time in terms of prior distribution, on top of a full-blown likelihood function that is more often than not just a big nuisance parameter), but I think the paper covers this aspect just too lightly. I think it's of benefit to the authors to release a published version of this paper once they have some more formal guidance or a more complex experimental setup providing a more thorough insight of it. I do think the contribution is really promising, but it feels unfinished, and I'd be curious to see where it could go following this direction.